# Deep Learning-Based Phenological Event Modeling for Classification of Crops

**Pattathal V. Arun** [1] and **Arnon Karnieli** [2,*]

1 Swiss Institute for Dryland Environmental and Energy Research, Jacob Blaustein Institutes for Desert Research, Sede Boker Campus, Ben Gurion University of the Negev, Midreshet Ben-Gurion 8499000, Israel; arun@post.bgu.ac.il

2 The Remote Sensing Laboratory, French Associates Institute for Agriculture and Biotechnology of Dryland, Jacob Blaustein Institutes for Desert Research, Sede Boker Campus, Ben-Gurion University of the Negev, Midreshet Ben-Gurion 8499000, Israel

* Correspondence: karnieli@bgu.ac.il; Tel.: +972-52-8795925

**Abstract:** Classification of crops using time-series vegetation index (VI) curves requires appropriate modeling of phenological events and their characteristics. The current study explores the use of capsules, a group of neurons having an activation vector, to learn the characteristic features of the phenological curves. In addition, joint optimization of denoising and classification is adopted to improve the generalizability of the approach and to make it resilient to noise. The proposed approach employs reconstruction loss as a regularizer for classification, whereas the crop-type label is used as prior information for denoising. The activity vector of the class capsule is applied to sample the latent space conditioned on the cell state of a Long Short-Term Memory (LSTM) that integrates the sequences of the phenological events. Learning of significant phenological characteristics is facilitated by adversarial variational encoding in conjunction with constraints to regulate latent representations and embed label information. The proposed architecture, called the variational capsule network (VCapsNet), significantly improves the classification and denoising results. The performance of VCapsNet can be attributed to the suitable modeling of phenological events and the resilience to outliers and noise. The maxpooling-based capsule implementation yields better results, particularly with limited training samples, compared to the conventional implementations. In addition to the confusion matrix-based accuracy measures, this study illustrates the use of interpretability-based evaluation measures. Moreover, the proposed approach is less sensitive to noise and yields good results, even at shallower depths, compared to the main existing approaches. The performance of VCapsNet in accurately classifying wheat and barley crops indicates that the approach addresses the issues in crop-type classification. The approach is generic and effectively models the crop-specific phenological features and events. The interpretability-based evaluation measures further indicate that the approach successfully identifies the crop transitions, in addition to the planting, heading, and harvesting dates. Due to its effectiveness in crop-type classification, the proposed approach is applicable to acreage estimation and other applications in different scales.

**Keywords:** deep learning; phenological curves; VENμS; classification; denoising





## 1. Introduction

Crops in a particular environment have specific phenological stages at defined time intervals in the season [1,2]. Modeling periodic events in the life cycle of crops is essential for distinguishing these crops. The derived information forms the basis for decision making in various irrigation scheduling activities to evaluate crop productivity [2,3]. Phenology-based analyses aim to track the change of phenological trajectories that vary from one crop to another in terms of the start, duration, and occurrences of crop events [2–5]. Although single-date satellite images have been widely employed for crop-type mapping, the tradeoff between spatial and spectral resolution of satellite sensors and the spectral similarities

between the crops result in the misclassification of crop types [6]. Hence, phenology-based metrics have been employed for crop type mapping tasks to overcome the issues related to conventional crop classification methods [7]. A noticeable number of studies in agriculture focus on the extraction of phenological features using remotely sensed data [2,3,8]. Most recent studies use vegetation index (VI) time-series derived from multi-temporal remote sensing data to determine specific phenological events [9,10].

Different conventional classifiers have been employed to classify the time-series VI data [11–13]. Most of these studies illustrate the need to consider the specific nature of the data. Maselli et al. [14] employed a semi-empirical approach using multi-temporal meteorological data and normalized differential vegetation index (NDVI) images to estimate actual evapotranspiration. To address the issue of the effect of mixed pixels in crop area estimation, Pan et al. [15] proposed a crop proportion phenology index to express the quantitative relationship between the VI time-series and winter wheat crop area. Zhang et al. [8] integrated crop phenological information from the MODerate resolution Imaging Spectroradiometer (MODIS) to estimate the maize cultivated area over a large scale. Gumma et al. [16] used MODIS data to map the spatial distribution of the seasonal rice crop extent and area in a related work. Similar work by Kontgis et al. [16] highlights the importance of considering flooded and cloud-covered scenes within the dense time stacks of data to achieve effective mapping of seasonal rice cropland extents. Although the lengths and timings of different phenological events provide the distinguishable signatures for different crop types, the variations in these characteristics due to different plant-, environment-, and sensor-related constraints may affect the effectiveness of phenology-based crop fingerprint estimation [2,3,17]. Hence, there is a need to derive the most important events from the data to distinguish different crops while resolving the issues of modeling errors. Some phenology-based classification approaches [9,18–22] have shown that mapping efficiency can be improved by adding important features that lead to better discrimination between the crop types. In this regard, phase and amplitude information derived using the Fourier transformation (FT) of the time-series data are employed to describe the vegetation status over time [9,19–21]. In addition to the Fourier-based harmonic analysis, thresholding and moving average of VI curves [23–26], slope and valley point analysis of the VI curves [27–29], and curvature-change rate analysis of logistic vegetation growth models [10,24,30–32] are applied to detect phenological events in the time-series remote sensing data [9]. However, most of these approaches require manual fine-tuning, are either supervised or semi-supervised in nature, and are sensitive to noise.

Deep learning (DL) approaches, which learn abstract representations to transform inputs into intrinsic manifolds in an unsupervised manner, have reported better results than the conventional machine learning approaches for various Earth observation (EO) data applications [33,34]. Variational autoencoders (VAEs) [35,36] learn the latent space as composed of a mixture of distributions enabling latent variable disentanglement and facilitate interpretability. Wang et al. [37] adopted an adversarial training process to adapt VAEs to model the inherent features of the spectral information effectively. Although convolutional neural networks (CNNs) have illustrated the capability to generate task-specific features, the handling of sensor limitations and acquisition errors requires these networks to have flexibility in defining the receptive fields [38]. The generative adversarial network (GAN)-based approaches applicable for the classification of VI curves generally use a one-to-one correlation-based similarity measure and are prone to shifts and distortions prevalent in the VI curves [33,39,40]. Long Short-Term Memory (LSTM)-based approaches adopt a recurrent guided architecture to model the sequential patterns. However, most of the existing DL classifiers, including LSTMs and GANs, consider the spectral curves as vectors that ignore the characteristics of physically significant features [41]. Although dynamic time wrapping (DTW)-based approaches consider the shape similarity and shifting in VI curves, the parameter tuning requirements affect their effectiveness and generalizability [42]. A recent advancement in CNN, called capsule networks, adopts capsules (a group of neurons) to address the issues of translation invariance prevalent in conventional CNNs [41,43,44].

Shi et al. [45] employed capsule blocks to model the spectral–spatial features to achieve high accuracy and interpretability in HSI-based classification tasks. Similar research has also been reported in [41,46–49], in which capsule networks were explored for EO data classification. However, few studies have reported the use of capsules for time-series classification [50,51].

Phenology-based index curve classification requires that VI curves are denoised to produce a smooth time-series [52]. Different algorithms, such as iterative weighted moving filters [53–57], nonlinear curve fitting [57–62], filtering in the Fourier domain [63,64], and spline-based smoothing [52,65,66], are being widely used for VI curve smoothing. However, most of these approaches either do not consider the phenological events, or require manual fine-tuning to avoid extraneous oscillations and to consider the specific nature of the phenological index curves. Although DL-based denoising approaches learn nonlinear feature spaces to avoid linear events, sparsity, and low-rank assumptions of the traditional interpolation methods [67–69], they generally do not consider the irregular sampling of the data and phenological events [70]. The DL-based approaches that have attained success in processing irregularly distributed point data [38,70,71] are not directly applicable to denoising VI curves. Moreover, the existing phenology-based classification approaches consider denoising, data imputation, outlier elimination, and classification as independent problems.

In this research, we hypothesize that capsule-based feature learning can adequately model the characteristic features of the VI curves and the crop-specific phenological events, such as growth transitions, planting, heading, and harvesting. The DTW-based neural units and interpolated convolution are hypothesized to dynamically learn kernels for estimating feature-specific shape similarity correspondences of the VI curves. It is also proposed that the joint optimization of denoising, data imputation, outlier elimination, and classification stages yields better results than the conventional approach of independently optimizing them. In addition, variational encoding conditioned with time-series aggregation is hypothesized to facilitate the consideration of the intra-crop phenological event variations and resolution of outliers. The current study also verified that the high measurement accuracy results from an appropriate latent representation and not from the exploitation of artifacts in the data [72–74]. The main contributions of the study are: (1) an interpretable VI curve classifier is proposed to consider the specific characteristic phenological events and the available prior knowledge; and (2) denoising, imputation, and classification stages are jointly optimized, considering the modeling errors and outlier effects, with a minimal number of training samples.

## 2. Materials and Methods

### 2.1. Datasets

The current study employed the Vegetation and Environment monitoring New Micro-Satellite (VENµS) data collected over three agricultural farms in Israel for phenology-based crop classification. The VENµS sensor is characterized by a high spatial resolution of 5 m, a high spectral resolution of 12 narrow bands in the visible to near-infrared regions of the spectrum, and a high revisit time of 2 days at the same viewing and azimuth angles. The NDVI images derived from the multi-temporal VENµS images of barley and wheat crop fields were used for various analyses. The study area consists of three farms covering 16.1, 13.9, and 2.2 sq. km, as shown in Figure 1. Among these, 40 fields of barley and 90 fields of wheat were considered in this research. The crops were sown during the first week of December and were harvested towards the end of April of the crop calendar years. Analysis of the proposed frameworks was conducted for the crop years 2017–2018, 2018–2019, and 2019–2020.

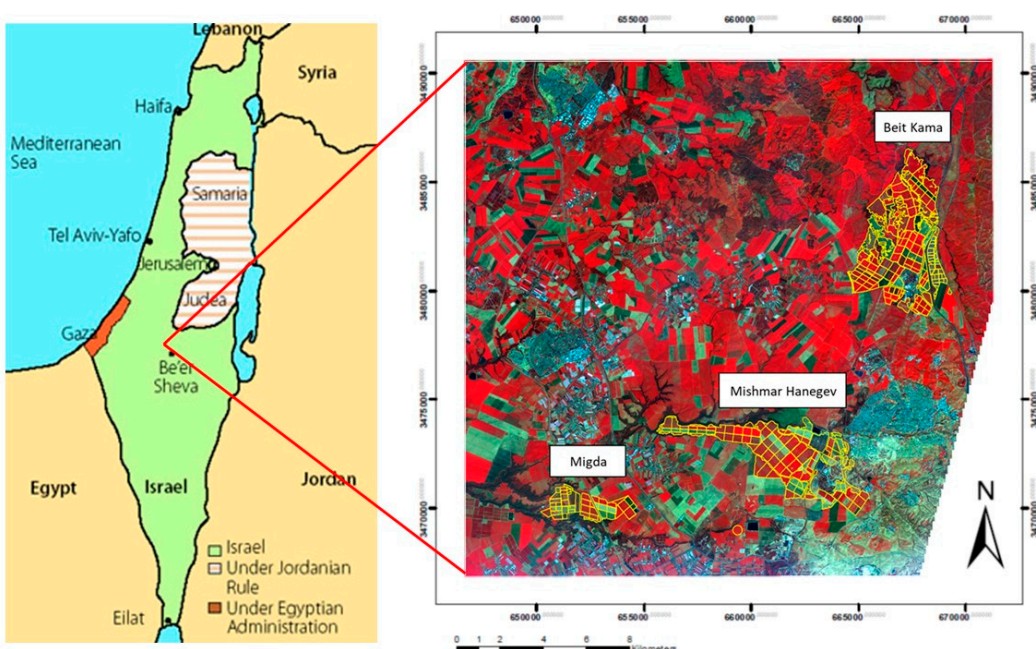

**Figure 1.** Map of Israel and a VENμS image showing the location of the experimental fields.

## 2.2. Proposed Approach

This subsection discusses the proposed variational capsule network, called VCapsNet, presented in Figure 2. The input VI curve is fed to both the capsule and LSTM streams. The capsules [43] are a group of neurons that model characteristics of features as the orientation of their output vectors. The proposed capsule stream constitutes 1D primary and class capsules representing the features and the crop classes. The output vector of the class capsules is composed of the mean and standard deviation vectors. The outputs of class capsules, having the maximum mean vector length, are used to sample the latent code that is used to predict the label and reconstruct the index curve. It may be noted that the final cell state of the LSTM stream is employed as a conditional parameter for the sampling process.

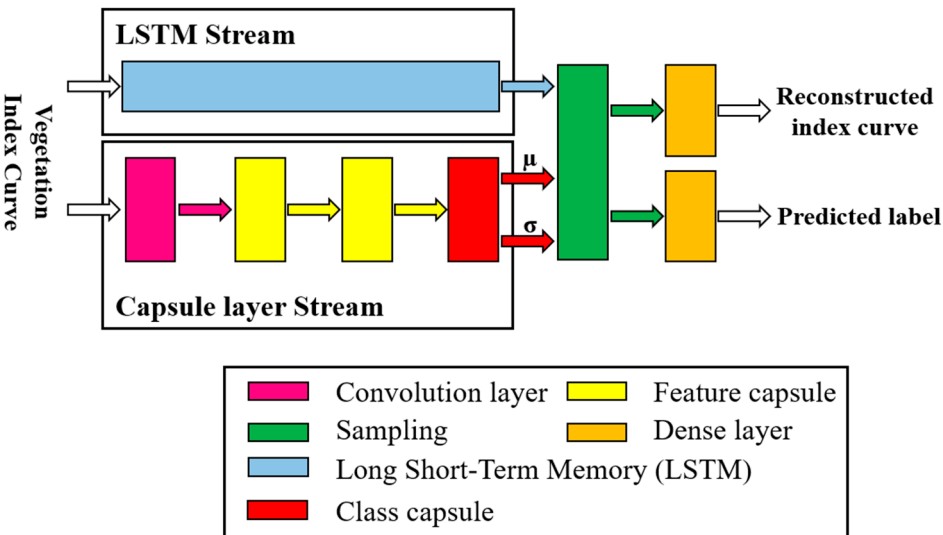

**Figure 2.** Proposed variational capsule network (VCapsNet) architecture without pooling (μ and σ denote mean and standard deviation, respectively).

### 2.2.1. Deep Capsule Network Stream

Unlike conventional capsule networks [43,44], this study proposes a deep 1D capsule network (Figure 2) to model the characteristic events and features of the index curves. The convolutional features of each layer of the capsule stream are squashed and reshaped to capsule form to be fed to the subsequent layers. In addition to stacking the convolutional layers in capsule form, another architecture (Figure 3) is also proposed, in which pooling layers are employed to model the events and features at different hierarchy levels. In both of the VCapsNet architectures, the feature tensors from the lower-level capsules are connected to each of the higher-level capsules. The length of the output of each capsule denotes the likelihood of finding the corresponding feature/pattern, and their orientation denotes the instantiation parameters.

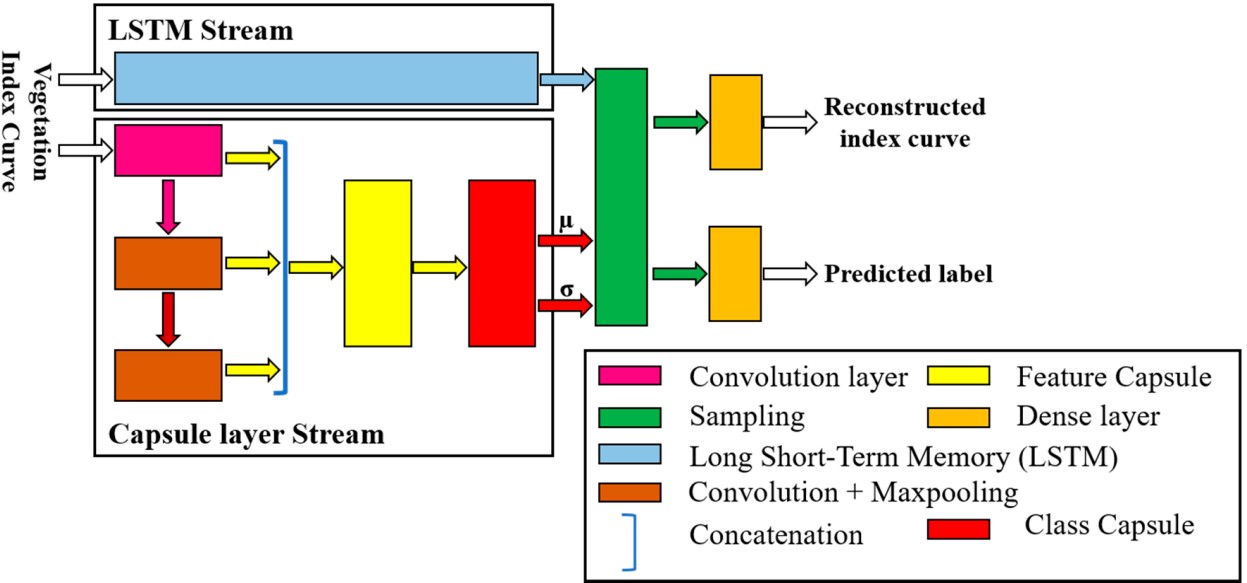

**Figure 3.** Proposed variational capsule network (VCapsNet) architecture with pooling (μ and σ denote mean and standard deviation respectively).

VCapsNet, as proposed in the current study, employs DTW-based nonlinear convolutional layers to resolve the issues of minor shifts in phenological events of the index curves of the same crop. The proposed DTW-based units facilitate one-to-many matchings of the receptive fields to better match the shapes of the index curve features. The DTW units match similar features to the input and skip elements with a considerable distance. The activation of a given DTW node n is computed as:

$$a_n = \phi \left( \sum_{(i,j) \epsilon S} ||w_{n,i} - v_j|| \right) \tag{1}$$

where $||.||$ is the L2 norm, v is the input vector of length m, and $w_n$ is the corresponding weight vector to the node n. The function $\phi(.)$ is a nonlinear activation function applied to the result, and S is a set of all the matched pairs between v and $w_n$ computed using the dynamic wrapping approach discussed in [42]. Aligning of network weights using DTW is repeated for every stride of the convolution during all forward passes. Consequently, the alignment is only maintained for the immediate forward and backward round, and recalculated on the fly for subsequent iterations. Further to modifying the neural units to implement one-to-many matching for considering the minor variations in phenological events, a modified form of the convolution is proposed to consider the irregularities in the

index curves. The interpolation-based convolution of vectorized VI curve v with a kernel $\kappa(.)$, centered at a location $\widetilde{x}$, is implemented as:

$$v * \kappa(\widetilde{x}) = \sum_{x'} \frac{1}{N_{x'}} \sum_{k_\alpha} \varphi(\kappa_\alpha, x') v(\widetilde{x} + x_\alpha).\kappa(x') \tag{2}$$

where $\varphi(.)$ is an interpolation function (e.g., Gaussian interpolation) that computes the weights based on a filter weight vector $k_\alpha$ and a given input point $x'$, and $N_{x'}$ is the density normalization term to make the convolutions sparsity invariant. In addition to conventional hyper-parameters, interpolation-based convolution uses the kernel length, defined as the distance between two adjacent weight vectors, to control the receptive field.

In the proposed VCapsNet architectures, the output $(u_i)$ of the ith capsule of layer l is transformed to obtain the prediction vector $\hat{u}_{j/i}$ of the jth capsule of the $l + 1$th layer as:

$$\hat{u}_{j/i} = T_{ij} u_i \tag{3}$$

where $T_{ij}$ is the transformation matrix between the ith and jth capsules of layers l and $l + 1$. The length of the output vector of the class capsule denotes the membership of the given index curve to the corresponding class. A nonlinear squashing function is employed to ensure that the short vectors are shrunk to almost zero length and long vectors are shrunk to a length slightly below 1. The squash function normalizes the magnitude of vectors rather than the scalar elements themselves. Hence, the output of the jth spectral class capsule $(v_j)$ is computed as:

$$v_j = \frac{||S_j||}{1 + ||S_j||} \frac{S_j}{||S_j||} \tag{4}$$

where $||.||$ denotes the L2 norm, and $S_j$ is the total input that is computed as the weighted sum of the prediction vectors $(\hat{u}_{j/i})$ of the capsules that are connected to the jth class capsule, i.e.,

$$S_j = \sum_i c_{ij} \hat{u}_{j/i} \tag{5}$$

The coupling coefficients $(c_{ij})$ between the ith and jth capsules are determined by a SoftMax routing procedure as:

$$c_{ij} = \frac{\exp(b_{ij})}{\sum_k \exp(b_{ik})} \tag{6}$$

The log priors $(b_{ij})$ are learned in addition to with the network weights and are iteratively refined by measuring the agreement between $v_j$ and $\hat{u}_{j/i}$. However, unlike in the conventional implementations, a DTW-based similarity measure is employed to consider the shape of the index curve features, i.e.,

$$b_{ij}^k = b_{ij}^{k-1} + \sum_{m,n \in S} \left|\left| v_j^{k-1}{}_m - \hat{u}_{j/i}^{k-1}{}_n \right|\right| \tag{7}$$

where $b_{ij}^k$ denotes the logits at each iteration k, $||.||$ denotes the L2 norm, S is the set containing the indices of elements along the warping path between the $v_j^{k-1}$ and $\hat{u}_{j/i}^{k-1}$. The wrapping path between two vectors is computed according to the discussions in [42,75]. It should be noted that the initial value of $b_{ij}$ $(b_{ij}^0)$ is the log prior probability that the i[th] capsule should be coupled to the j[th] capsule.

### 2.2.2. Conditional Variational Encoding

The mean $(\mu)$ and standard deviation $(\sigma)$ vectors of the class capsule having the maximum length are employed to sample the latent space through a reparameterization trick [35,76]. The final cell state learned by the LSTM, which integrates the sequential events of the phenological curves, is used to condition the sampling process. The sampled

latent code is used to reconstruct the index curve and predict the label using two separate dense layers. The proposed variational encoding accommodates the natural variations in the index curves of the same crop and improves the generalization capability of the network. In addition, pre-training of the LSTM using noisy samples ensures the approach is resilient to noise and other irregularities. The variational encoding loss is composed of a fitting data term (maximizing the data likelihood) and a latent compression term (ensuring that the latent code distribution $Q(z|v,h)$ stays close to the latent prior $P(v|z,h)$) as:

$$L_{VAE} = E(\log P(v|z,h)) - \beta D_{KL}(Q(z|v,h)||P(z|h)) \tag{8}$$

where $E(.)$ denotes an expectation value, P and Q are probability distributions, $D_{KL}(.||.)$ is the Kullback–Leibler divergence, v and z indicate the data and latent spaces, respectively, $\beta$ is the entanglement penalty factor, and h denotes the condition vector. The condition vector h is directly involved in the encoding and decoding processes, and the final cell state of the LSTM is used as the condition vector. Although the penalizing $D_{KL}(Q(z|v,h)||P(z|h))$ term in Equation (8) facilitates disentanglement, it results in the loss of information about the input v stored in the latent code z and, therefore, a poor reconstruction for high values of $\beta$. Hence, in this study, inspired by [77], the formulation in Equation (8) is modified as:

$$L_{VAE} = E\left(E_{q(z|v)}(-\log(P(v|z,h)))\right) - E(D_{KL}(Q(z|v,h)||Q(z))) - \beta D_{KL}\left(Q(z)||\prod_{j=1}^{d}Q(z_j)\right) \tag{9}$$

The last term in Equation (9) measures the dependence for multiple variables.

### 2.2.3. Loss Functions and Regularizations

In this study, the reconstruction and cross-entropy losses are minimized to train the network weights in accordance with the classification objective. To facilitate denoising in conjunction with classification, the spectral-dissimilarity- and DTW-based reconstruction losses are also employed as:

$$L_R = \frac{1}{m\pi}\sum_{i=1}^{m}\arccos\left(\frac{v.\tilde{v}}{|v||\tilde{v}|}\right) + \psi_\alpha(\langle A, \Delta(v,\tilde{v})\rangle) \tag{10}$$

where m is the length of the input pixel spectra v, $\tilde{v}$ is the reconstructed pixel spectra, $|.|$ denotes the L1 norm, $\psi_\alpha(.)$ is the generalized minimizing function with a smoothing parameter $\alpha$, $\Delta(.,.)$ denotes the cost matrix, and A is the alignment matrix. For training the network, in addition to the L2 regularization loss, to ensure piece-wise similarity, a multiscale version of the structural dissimilarity loss is also employed as:

$$L_{SD} = \sum_{p\in P}1 - \Omega(p) \tag{11}$$

where:

$$\Omega(p) = \frac{2\mu_p\mu_p' + C_1}{\mu_p^2 + \mu_p'^2 + C_1}\cdot\frac{2\sigma_p\sigma_p' + C_2}{\sigma_p^2 + \sigma_p'^2 + C_2} \tag{12}$$

where $P \subseteq R$ denotes the set of all relative locations on the VI curve, $C_1$ and $C_2$ are constants, $\mu_p$ and $\mu_{p'}$ respectively, represent the means of the patches of the reconstructed and ground-truth VI curves, and $\sigma_p$ and $\sigma_{p'}$ respectively, denote the corresponding standard deviations. The means and standard deviations are computed in neighborhoods (context) of varied extents to implement multiscale measurements of structural dissimilarity.

The reconstruction loss of VCapsNet is constrained to fine-tune the latent representations z for the classification task as:

$$L_{CR} = \alpha\sum_{i=1}^{m} -[y_i\log(\chi(w_e,z_i)) + (1-y_i)(1-\log(\chi(w_e,z_i)))] + \lambda|z_i| \tag{13}$$

where $y_i$ is the label for the ith sample, $\chi(.)$ corresponds to the normalization of the length of the class capsule's outputs, $w_e$ is the weight matrix of the encoding layers, $\alpha$ and $\lambda$ are the scaling factors, $z_i$ is the latent representation of the ith sample, $|.|$ denotes the L1 norm, and m is the number of samples. An additional classification loss is employed to incorporate the label information of the source domain into the embedding space as:

$$L_{CE} = -\frac{1}{n}\sum_{i=1}^{n}\sum_{j=1}^{c} I_{p_i=y_j} \log \frac{e^{p_i}}{\sum_{r=1}^{c} e^{p_{ir}}} \qquad (14)$$

where I is an indicator function, $y_j$ is the expected output of the ith sample corresponding to the jth class, $p_i$ is the predicted label for the ith sample, n is the number of samples, and c denotes the total number of classes. The indicator function $I_{p_i = y_j}$ outputs one when the predicted label ($p_i$) matches the expected label $y_j$.

### 2.2.4. Transparency and Interpretability

This Section attempts to understand the concepts/prototypes learned by the network and how the input features contribute to a given decision. These approaches are used to compare the DL models in terms of their interpretability and the physical significance of the learned features. The training data and hyper-parameters are also refined based on the analysis of learned prototypes and relevance score assignments. In the current study, the crop-specific characteristic phenological events, such as planting, heading, harvesting, and growth transitions, serve as the interpretability parameters; that is, the timing of different phenological events is the physically interpretable feature that distinguishes a crop type from others.

Inspired by [74,78,79], to interpret the proposed VCapsNet, an approach based on activation maximization was employed. The representative/prototype of the class $\omega_c$, which corresponds to the most likely input x for class $w_c$, is found by optimizing:

$$\max_{x}(\log p(\omega_c|x) + \log p(x)) \qquad (15)$$

where x is the input, and $p(\omega_c|x)$ and $p(x)$ are the class conditioned data density and data model, respectively. Although interpreting the concepts/prototypes learned by DL models helps compare and contrast different models, learning the contribution of input features for each concept is also equally important. A modified version of layer-wise relevance propagation (LRP) was employed to assign quantitative values to input features based on their relative significance in the output prediction. As shown in [80], the relevance can be distributed to input-layers using local redistribution rules as:

$$R_j^{(l-1)} = \sum_i \frac{a_i^{(l-1)} w_{ij}^{(l-1,l)}}{\sum_k a_k^{(l-1)} w_{kj}^{(l-1,l)}} R_i^{(l)} \qquad (16)$$

where $w_{ij}$ denotes the weight between neurons i and j, $a_i^{(l-1)}$ denotes the activation, and i indexes all neurons of layer l joined to neuron j. Equation (16) is applied backward through the network from the output layer to produce the relevance map. It should be noted that the summation of the relevance at any layer is conserved in the network. By analyzing the relevance scores, regions or patterns in the inputs that mainly contribute to a classification decision are identified. As the probability of an input belonging to a specific class depends on the value at the output layer neuron, relevance scores can represent evidence for (positive values) and against (negative values) the classification decision.

### 2.3. Implementation of the Proposed VCapsNet

This subsection discusses the implementation of the proposed VCapsNet model in the classification of VI index curves. The NDVI curves derived from multi-date VENμS images

were used to train and validate the network. The cloud masks were used to eliminate noisy pixels for NDVI computation. The shapefiles of crop fields, and the cropping and harvesting information obtained from farmers, served as ancillary data for labeling and analyzing the VI curves. It is noteworthy that the temporal index curves, having a vector length of 36, were used as inputs for analyzing the models. Data augmentation techniques, similar to those adopted in [81,82], were used to increase the number of training samples. In addition to other systematic noise simulations [53,56,83,84], a random approach was employed to remove values or add Gaussian noise at irregular intervals to train the models to facilitate denoising. Different downscaling strategies, such as bilinear, bi-cubic, and nearest neighbor interpolations, were employed for a specific number of VI curves to generate training and testing samples for data imputation. Multiple downscaling strategies were adopted to avoid the bias of the trained network towards a particular approach.

The VCapsNet implementation adopted in this study uses multi-sized kernels of sizes $1 \times 2$, $1 \times 3$, $1 \times 5$, $1 \times 7$, and $1 \times 9$ in the capsule streams of architectures-1 and -2. ReLU activations follow the padded convolutions in all layers. The number of filters and the stride of pooling layers are respectively set to 256 and 2 for the capsule stream of architecture-1 (Figure 2). For the capsule stream of architecture-2 (Figure 3), the number of filters is set to 256, 128, and 64. As the input vectors are of length 36, the LSTM stream employed in both architectures uses 36 layers. The variational encoding is implemented as an encoding-decoding architecture consisting of four convolutional and deconvolutional units and a fully connected sampling layer. The convolutional units are followed by maxpooling, whereas deconvolution units are preceded by upsampling layers. The number of filters in the eight-layer deep variational stream is set to 256, 192, 128, 64, 64, 128, 192, and 256. Gaussian interpolation is used in all of the architectures as the convolutional interpolation function, and the Gaussian bandwidth ($3\sigma$) is fixed to 0.1. It is important to note that, for implementing multiscale structural dissimilarity measurements, the context extents are varied among 1, 3, 5, 7, and 9.

In the current implementation, rather than using a standard dropout algorithm [85], which may change the properties of the entity that the capsule represents, the approach discussed in [86] was adopted. The mean squared error (MSE)-based loss and cosine dissimilarity loss [87] and the proposed losses are minimized to learn the network weights. Hyper-parameter optimization, proposed in [88], is employed to optimize the hyper-parameters, such as kernel size, number of filters, depth of the network, and number of epochs. Architecture-1 was trained using ADAM optimization for 300 epochs with an initial learning rate of 0.01 and a decay rate of 0.5 for every 100 epochs. Similarly, architecture-2 was trained using ADAM optimization for 250 epochs with an initial learning rate of 0.008 and a decay rate of 0.5 for every 100 epochs. A batch size of 50 was employed for training of both networks.

## 3. Results

To verify the effectiveness of VCapsNet, extensive experiments were conducted for phenological curve-based crop classification. The pixel-level NDVI curves derived from multi-date images were used to train VCapsNet to distinguish different types of crops. The ancillary data was used to assign labels to ground-truth phenological curves. Data augmentation, similar to that adopted in [81,82], was used to increase the number of training samples. In addition, random Gaussian noise was added in irregular intervals to evaluate the effect of denoising. An example of augmented patterns for the barley crop is presented in Figure 4. VCapsNet was extensively analyzed using the data of three farms over three consecutive crop years. A total of 4600 samples were used for training and testing the model, among which 800 were augmented patterns. Hyper-parameter optimization, proposed in [88], was employed to optimize the parameters of the different models analyzed in this study. It should be noted that an early stopping framework using k-fold validation forms the basis of the parameter selection. For all of the experiments adopted, k-fold validation was used with k set to 10. The confusion-matrix-based Kappa

statistic and overall accuracy were used for evaluating the classification results. High values of Kappa statistic and overall accuracy indicate high accuracy. A Z-score-based test statistic (discussed in [89,90]) was employed to analyze the significance of the results presented in the current study. In addition to confusion-matrix-based measures, proposed interpretability techniques (Section 2.2.4) were used to evaluate the physical significance and interpretability of the models. The peak signal-to-noise ratio (PSNR) was used to estimate the denoising accuracy of VCapsNet and other benchmark denoising approaches. The ablation analysis of VCapsNet is discussed in Section 3.1. A comparative analysis of VCapsNet with the benchmark approaches is presented in Section 3.2.

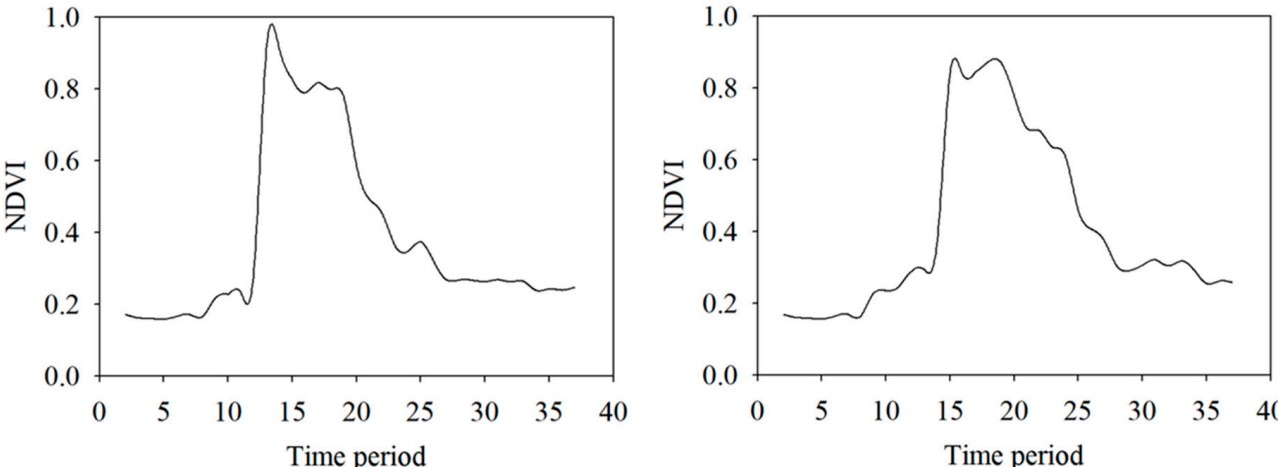

**Figure 4.** Illustration of augmentation for the vegetation index curve of the barley crop.

### 3.1. Ablation Analysis of VCapsNet

This Section evaluates the effect of the proposed architectural variations and loss functions on the results. The results are summarized in Table 1. It is observed that the proposed strategies reduce the training sample requirement and significantly improve the results (in terms of Kappa and overall accuracy). As is evident from Table 1, the implementation of VCapsNet without a capsule stream results in lower values of Kappa and overall accuracy. These results illustrate that the capsules model the phenological events and their characteristic features effectively to distinguish different classes. Moreover, the use of capsules significantly reduces the network depth. It may be noted from Table 1 that the implementation of VCapsNet without a variational encoding strategy results in lower Kappa and overall accuracy values. The improvement in accuracy due to the variational encoding strategy can be attributed to the consideration of minor phenology variations in the same crop type and to the resolution of outliers. The use of DTW-based routing technique also improves the results by facilitating the consideration of shapes of the phenological events. In addition, as is observed from the Kappa and overall accuracy values in Table 1, the use of the cell state learned using LSTM for conditioning the latent space sampling acts as a regularizer. This resolves the issues of vanishing gradient and convergence. The use of reconstruction loss as a means to regulate the classification loss facilitates denoising and data imputation of the index curves. Training of the proposed network for denoising and imputation ensures the classification is resilient to noise and other irregularities. Similarly, the use of piece-wise loss, interpolation-based convolution, and DTW-based neural units improves the classification accuracy. The proposed architectures and losses improve the modeling of phenological events that are evident from the concepts learned for different types of crops. The relevance analysis of input features indicates that VCapsNet focuses on features and phenological events that are physically significant to each crop type. Additionally, fine-tuning of the entanglement penalty facilitates the disentanglement of the latent codes concerning different crop classes.

**Table 1.** Analysis of the effect of the proposed architectures and constraints for 50% of the training samples *.

| Architectural Variations/Losses | Kappa Statistic | Overall Accuracy | Z-Score |
|---|---|---|---|
| 1D Capsule based classifier | 0.81 | 83.46 | 2.21 |
| Long Short-Term Memory (LSTM) based classifier | 0.82 | 85.91 | 2.18 |
| VCapsNet without Capsule Stream | 0.85 | 87.18 | 2.25 |
| VCapsNet without Long Short-Term Memory (LSTM) stream | 0.86 | 89.68 | 1.99 |
| VCapsNet without variational encoding | 0.86 | 91.24 | 2.32 |
| VCapsNet without embedding the label information prior | 0.86 | 92.68 | 2.16 |
| VCapsNet without fine-tuning the latent space for classification | 0.86 | 90.12 | 2.17 |
| VCapsNet without piece-wise reconstruction loss | 0.87 | 88.56 | 2.09 |
| VCapsNet without cosine dissimilarity loss | 0.88 | 91.43 | 1.97 |
| VCapsNet without Dynamic Time Wrapping (DTW) loss | 0.87 | 89.08 | 1.99 |
| VCapsNet without interpolated convolution | 0.89 | 94.12 | 2.12 |
| **Proposed VCapsNet implementation** | **0.94** | **98.57** | **-** |

* Z-score > 1.96 shows a significant (>95%) difference between the confusion matrices of the existing approaches and the variational capsule network (VCapsNet).

A comparison of both of the proposed architectures indicates that, at a lower number of training samples, architecture-2 is preferred, whereas architecture-1 provides better results when enough training samples are available. In addition, architecture-2 yields acceptable results at much shallower depths compared to architecture-1.

The sensitivity analysis of the proposed VCapsNet models in terms of network parameters is presented in Figures 5 and 6. For a given set of training samples, an increase in the network depth improves the Kappa value that, however, deteriorates as the number of layers increases beyond a limit. Empirically, for input VI curves having a length of 24–36, a 5–9 layered network yields the best results. The increase in the number and sizes of filters improves the accuracy, which slowly saturates and deteriorates following further blind increase. The increase in size and number of filters exponentially increases the computational complexity of the network. As the length of phenological features can vary, the use of multi-sized kernels significantly improves the results without significantly affecting the execution time. The reduction in the sensitivity of VCapsNet to network parameters can be attributed to the effective modeling of characteristic features of index curves.

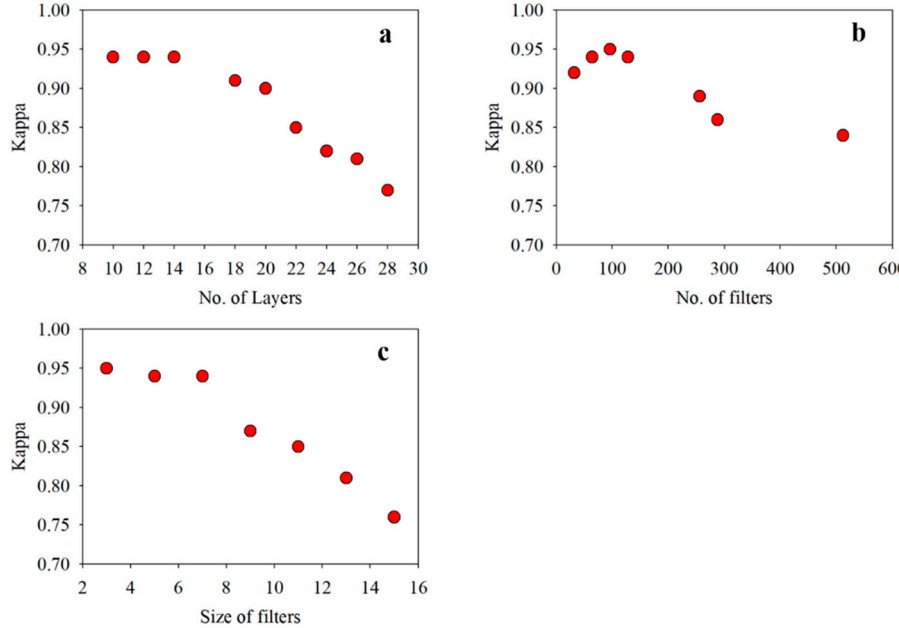

**Figure 5.** Sensitivity analysis of the variational capsule network (VCapsNet) in terms of (**a**) network depth; (**b**) size of filters; and (**c**) number of filters for classification.

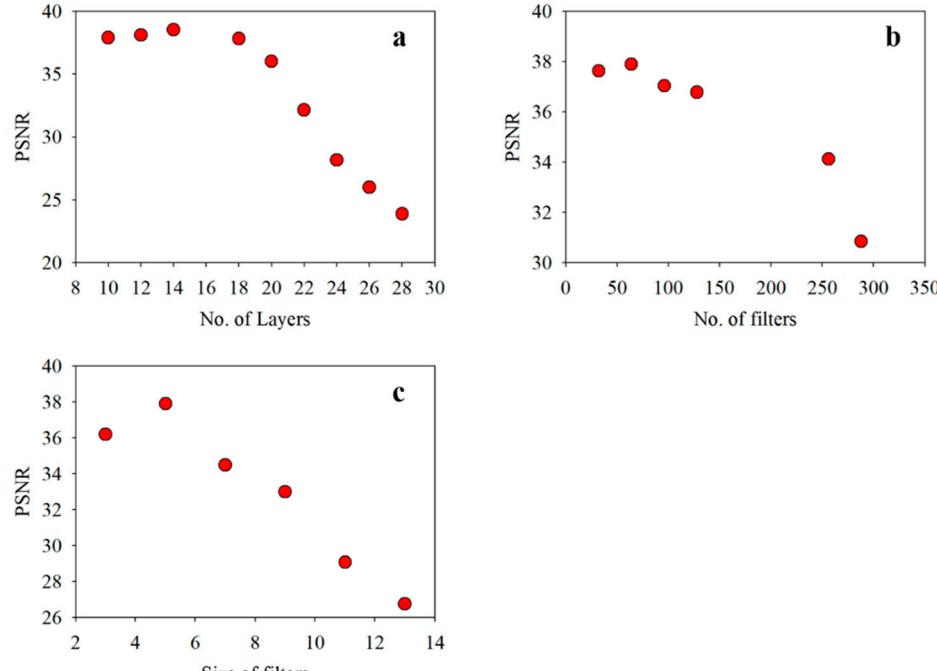

**Figure 6.** Sensitivity analysis of the variational capsule network (VCapsNet) in terms of (**a**) network depth; (**b**) size of filters; and (**c**) number of filters for denoising.

### 3.2. Comparison of VCapsNet with the Commonly-Used DL Based Approaches

The commonly-used classifiers applicable to VI curve classification were compared with the proposed VCapsNet-based approach. The results are summarized in Table 2. The significance of the results of VCapsNet (at a confidence level of 95%) in comparison with the other approaches is analyzed in Table 3. In all of the experiments adopted in this study, the entire set of 4600 samples was split into disjoint training and testing subsets. For instance, when the percentage of training samples is 10%, 90% of the samples were used for testing the model. Based on the discussions in [2,3,7,12,39,83,91], some main existing classification approaches were selected as the benchmark methods for comparison. In the experiments discussed in this study, a few of the benchmark approaches were modified to process the one-dimensional phenological curves. An analysis of the variation in accuracy of different approaches according to the variation in the percentage of training samples is presented in Figure 7. Furthermore, Figure 7 presents the variation in accuracies for different folds of k-fold validation in terms of the standard deviation. A total of 4600 training samples were used, and 10-fold validation was employed for each of the different sub-experiments (10%, 20%, 30%, etc.). As is evident from the results, VCapsNet better models phenological curves compared to other prominent approaches. The proper modeling of phenological events and features significantly improves the generalization capability of the network, resulting in improved classification accuracies even with a small number of training samples (Figure 7). The learning of physically significant features and phenological events also resolves the issues of intra-crop variability of the phenological curves. In addition, the DTW-based convolutional units and interpolation-based convolutions, and the proposed losses and regularizations, facilitate the effective transformation of vectorized phenological curves to a latent space that is more discriminative than the original space.

**Table 2.** Comparison of the variational capsule network (VCapsNet) with benchmark deep learning (DL) classifiers for 40% of training samples *.

| Benchmark Classifiers | Kappa Statistic | Overall Accuracy |
|---|---|---|
| Deep learning (DL) based [92] | 0.74 | 80.10 |
| Spectral attention CNN [93] | 0.79 | 84.43 |
| Phenology metrics based [9] | 0.85 | 87.91 |
| Bayesian estimator based [2] | 0.83 | 88.20 |
| Representation learning based [94] | 0.86 | 90.08 |
| Generative Adversarial Network (GAN) based [39] | 0.83 | 88.36 |
| Multivariate Long Short-Term Memory (LSTM) based [95] | 0.87 | 90.82 |
| **Proposed VCapsNet** | **0.92** | **96.23** |

* Z-score > 1.96 shows a significant (>95%) difference between the confusion matrices of the existing approaches and the variational capsule network (VCapsNet).

**Table 3.** Z-score-based significance analysis of variational capsule network (VCapsNet) in comparison with the benchmark deep learning (DL) classifiers *.

| Benchmark Smoothing Approaches | Z-score of Kappa Statistic as Compared to VCapsNet | Z-score of Overall Accuracy as Compared to VCapsNet |
|---|---|---|
| Deep learning (DL) based [92] | 2.03 | 1.98 |
| Spectral attention Convolutional Neural Network (CNN) [93] | 1.98 | 2.21 |
| Phenology metrics based [9] | 2.12 | 2.14 |
| Bayesian estimator based [2] | 2.64 | 2.43 |
| Representation learning based [94] | 2.07 | 2.07 |
| Generative Adversarial Network (GAN) based [39] | 2.59 | 2.46 |
| Multivariate Long Short-Term Memory (LSTM) based [95] | 2.18 | 1.99 |

* Z-score > 1.96 shows a significant (>95%) difference between the confusion matrices of the existing approaches and the variational capsule network (VCapsNet).

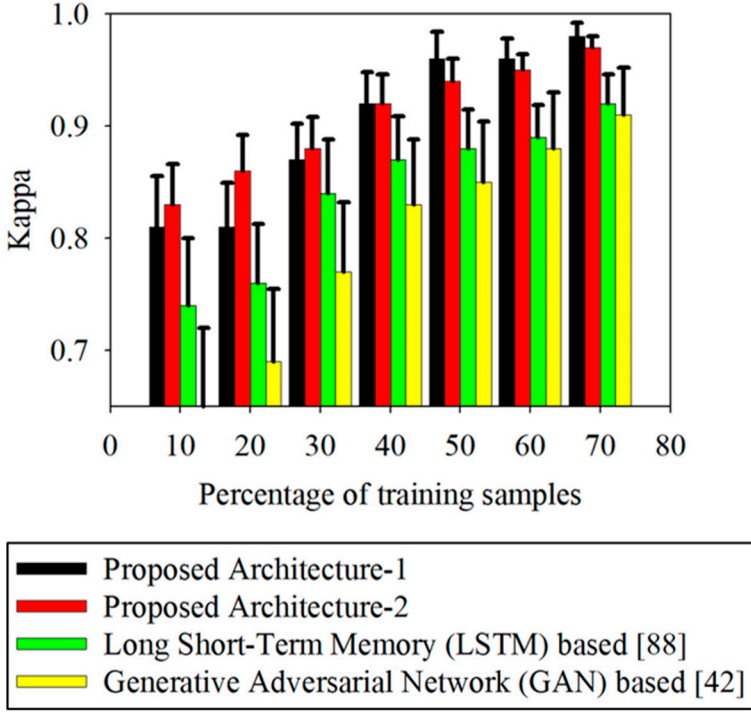

**Figure 7.** Comparison of classifiers with respect to varying percentages of training samples.

In addition to the classification-based accuracy assessment, the models were also evaluated based on the prototypes learned for each crop, as discussed in Section 2.2.4. For this purpose, the phenology curves that follow the correct timeline for each crop were selected based on the ancillary data and generalized using a variational autoencoder. The benchmark phenological curve for each crop, generated by sampling the mean of such a learned latent space, was compared with the learned concepts of the corresponding crops for each model. In this regard, cosine-based, DTW-based, and Fourier-based approaches were used as the similarity measures. The dates of crop-specific phenological events, such as growth transition, planting, heading, and harvesting, were adopted as characteristic features for comparison. A summary of these comparisons is presented in Table 4. The harmonics of the Fourier transform of the VI curves properly capture the phenological events, whereas DTW approaches measure the shape-similarity irrespective of the minor shifts. The high values of similarity measures indicate that VCapsNet accurately learns the phenological events and physically significant features. A visual illustration of the predicted and actual NDVIs corresponding to the crop-specific phenological events for randomly selected wheat and barley fields is presented in Figure 8. As is evident, VCapsNet provides accurate results.

**Table 4.** Interpretability based comparison of the different deep learning (DL) models.

| Benchmark Classifiers. | Normalized Cosine Similarity Between the Concepts Learned and the Benchmark Phenological Curves | Normalized RMSE in Fourier Domain Between the Concepts Learned and the Benchmark Phenological Curves | Normalized DTW Based Similarity Between the Concepts Learned and the Phenological Benchmark Curves |
|---|---|---|---|
| Deep learning (DL) based [92] | 0.9874 | 0.4290 | 0.7742 |
| Spectral attention Convolutional Neural Network (CNN) [93] | 0.9870 | 0.3916 | 0.7891 |
| Representation learning based [94] | 0.9942 | 0.2396 | 0.8263 |
| Generative Adversarial Network (GAN) based [39] | 0.9964 | 0.2246 | 0.8551 |
| Multivariate Long Short-Term Memory (LSTM) based [95] | 0.9968 | 0.2389 | 0.8629 |
| **Proposed VCapsNet** | **0.9989** | **0.1059** | **0.9672** |

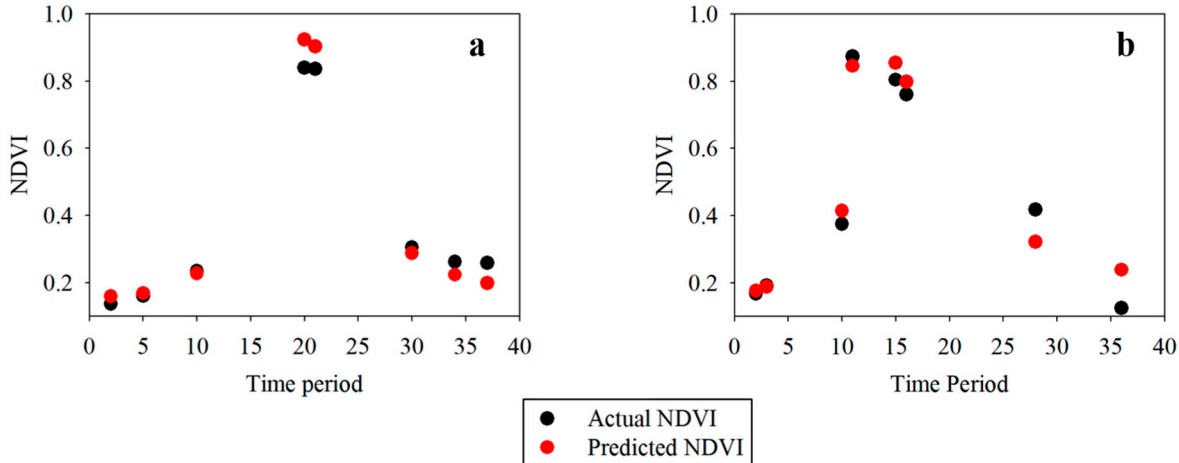

**Figure 8.** Expected and predicted NDVI corresponding to wheat (**a**) and barley fields (**b**).

To further explain the network and analyze the contribution of input features, the LRP approach (Section 2.2.4) was adopted. The propagated relevance of VCapsNet for

distinguishing different crop types indicates that the model places importance on the time frames related to phenological events.

To analyze the temporal effectiveness of VCapsNet, a leave-one-out validation strategy was adopted year-wise. The training samples of two crop years were used to classify the crop phenology from another crop year. The crop years of 2017–2018, 2018–2019, and 2019–2020 were considered for the analysis. A total of 3500 samples were used for training and 1500 samples for testing. The result of the experiment is presented in Table 5 and the significance of the analysis in Table 6. The performance and generalizability of the proposed approach can be attributed to the effective modeling of the characteristic phenological features of the crops.

**Table 5.** Comparison of the different deep learning (DL) models across different crop years.

| Benchmark Classifiers. | Overall Accuracy | Kappa Statistic | Normalized DTW Similarity Between the Concepts Learned and the Phenological Benchmark Curves |
|---|---|---|---|
| Deep learning (DL) based [92] | 67.41 | 0.62 | 0.6591 |
| Spectral attention Convolutional Neural Network (CNN) [93] | 71.82 | 0.65 | 0.6983 |
| Representation learning based [94] | 76.33 | 0.72 | 0.7150 |
| Generative Adversarial Network (GAN) based [39] | 74.49 | 0.66 | 0.6840 |
| Multivariate Long Short-Term Memory (LSTM) based [95] | 79.09 | 0.74 | 0.7651 |
| **Proposed VCapsNet** | **89.72** | **0.86** | **0.9439** |

**Table 6.** Z-score-based significance analysis of the temporal validation of VCapsNet *.

| Benchmark Smoothing Approaches | Z-Score of Kappa Statistic as Compared to VCapsNet | Z-Score of Overall Accuracy as Compared to VCapsNet |
|---|---|---|
| Deep learning (DL)-based [92] | 1.97 | 2.01 |
| Spectral attention Convolutional Neural Network (CNN) [93] | 2.17 | 2.34 |
| Phenology metric-based [9] | 2.09 | 2.16 |
| Bayesian estimator-based [2] | 1.99 | 2.05 |
| Representation learning-based [94] | 2.14 | 2.37 |
| Generative Adversarial Network (GAN)-based [39] | 2.08 | 2.42 |
| Multivariate Long Short-Term Memory (LSTM)-based [95] | 1.97 | 2.14 |

* Z-score > 1.96 shows a significant (>95%) difference between the confusion matrices of the existing approaches and the variational capsule network (VCapsNet).

### 3.3. Comparison of VCapsNet with the Commonly-Used Denoising Approaches

In addition to classification, VCapsNet reconstructs the phenological index curves, thereby facilitating denoising and data imputation. In this section, the commonly-used denoising approaches applicable to VI curve smoothing are compared with the proposed VCapsNet. In all of the experiments adopted in the study, the entire sample set was split into disjoint training and testing subsets. As the reconstruction is conditioned based on the crop type information, the proposed approach of joint classification and denoising provides better results than the existing denoising approaches. The selected benchmark approaches are improved versions of those that reported the state-of-the-art results in [52,54,63–65,96,97]. The result of the comparative analysis is summarized in Table 7 and Figure 9. Table 8 confirms the significance of the comparison at a confidence level of 95%. It is to be noted that Figure 9 presents the bar graphs indicating the standard deviation during the k-fold validation. The local filtering methods (Savitzky–Golay filtering and locally weighted scatterplot

smoothing) resulted in better performance with optimized parameter settings among the conventional smoothing approaches. However, fitting methods (asymmetric Gaussian function fitting and double logistic function fitting) are less sensitive to the parameters. As is evident from the results, VCapsNet yields higher PSNR (better reconstruction accuracy) than the benchmark approaches considered in this study. The results of VCapsNet can be attributed to its ability to model phenological events. In addition, the proposed variational encoding strategy, LSTM-based conditioning, and loss functions, significantly improve the generalizability. It should be noted that VCapsNet learns the smoothing parameters dynamically, whereas conventional smoothing approaches require manual fine-tuning. Additionally, the phenology event-based latent space resolves the issues of domain bias and inter-field variability of VI curves of the same crop.

**Table 7.** Comparison of VCapsNet with benchmark smoothing approaches for 50% of training samples *.

| Benchmark Smoothing Approaches | PSNR |
|---|---|
| Least squares fitting to double logistic functions [57] | 24.32 |
| Least square fitting to asymmetric Gaussian functions [62] | 27.92 |
| Spline smoothing [66] | 29.05 |
| Deep learning (DL) based [69] | 31.93 |
| Savitzky-Golay filter based [56] | 29.51 |
| Deep learning (DL) based [70] | 33.74 |
| Savitzky-Golay filter based [53] | 31.83 |
| **Proposed VCapsNet implementation** | **37.90** |

* Benchmark methods are implemented based on the available GitHub implementations and are fine-tuned based on the related publications.

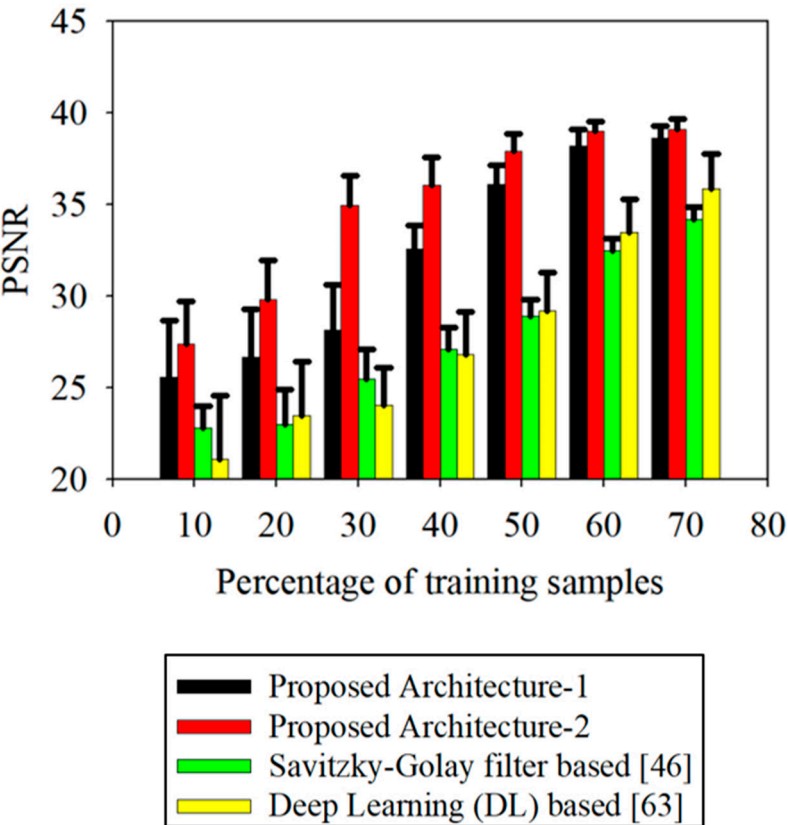

**Figure 9.** Accuracy analysis of smoothing approaches with respect to the change in the percentage of training samples.

**Table 8.** Z-score-based significance analysis of the results of VCapsNet *.

| Benchmark Smoothing Approaches | Z-Score with Respect to VCapsNet |
|---|---|
| Least squares fitting to double logistic functions [57] | 2.38 |
| Least square fitting to asymmetric gaussian functions [62] | 2.20 |
| Spline smoothing [66] | 2.09 |
| Deep learning (DL) based [69] | 1.99 |
| Savitzky-Golay filter based [56] | 2.29 |
| Deep learning (DL) based [70] | 2.20 |
| Savitzky-Golay filter based [53] | 2.16 |

* Z-score > 1.96 shows a significant (>95%) difference between the confusion matrices of the existing approaches and the variational capsule network (VCapsNet).

## 4. Discussion

Experiments on VCapsNet (discussed in Section 4) illustrate that the proposed approaches yield better results than the main existing approaches. A detailed analysis of the results is presented in the following subsections.

### 4.1. Modeling Phenological Events

Learning and appropriate modeling of different phenological events and their characteristics, such as relative locations, length, shape, texture, and context, are essential for crop classification. The architecture of DL models is found to influence the capability in modeling the phenological curves. The use of capsules enables better modeling of the relative locations and features of the phenological events compared to the convolutional networks. The proposed DTW-based routing mechanism facilitates the consideration of the contexts of phenological events for the effective modeling of VI curves. Although VCapsNet yields better results than the existing approaches in terms of the Kappa statistic and overall accuracy, these accuracy measures do not ensure that the network models the phenological curves correctly. In this regard, interpretability and explanation-based analyses must be able to indicate if the network has correctly learned the concepts for each class and confirm if it is appropriately placing importance on the input features. Among the two proposed architectures of VCapsNet, architecture-2 yields better results when the samples are limited, whereas architecture-1 performs better when sufficient training samples are available. The use of variational encoding for fitting the latent distribution to a normally distributed space resolves the issues of crop phenological variability and the problem of outliers. However, additional sampling layers in variational autoencoders are found to adversely affect the modeling of VI curves, especially when training samples are limited. It is further observed that the use of a LSTM cell state to condition the latent space regulates the sampling process, resulting in a feature space that captures the characteristic phenology of each crop. It may be noted that convolutional networks rather than capsules do not yield acceptable results due to the need to consider the specific characteristics of the VI curves. The LSTM classifiers only consider the sequential nature of the curves and ignore the characteristic features of the phenological events and their relative locations. Moreover, as adopted in conventional capsules, the simple routing ignores the shape similarity of the phenological events.

This study illustrates that the joint optimization of denoising, data imputation, and classification yields better results than individually optimizing them. The proposed VCapsNet provides not only good classification results, but is also effective in denoising and data imputation. In VCapsNet, the reconstruction serves as a regularizer for classification, and class information is a regularizer for denoising and data imputation. The proposed constraints and losses use the input priors to improve the projected space, thereby resulting in meaningful and interpretable representations that capture the phenological events and features. Different experiments in the current study illustrate that using multi-sized kernels facilitates the modeling of phenological features having variable lengths.

VCapsNet yields good results even at shallower depths compared to the other conventional DL approaches. The proposed regularizations and data augmentations, in addition

to capsule-based feature modeling, improve the generalizability of the network. VCapsNet also considers the inter-field variability of the crops through proper modeling of the phenological events and embedding the label information in the latent space. Moreover, the neural units, convolutions, and routing mechanisms of VCapsNet are modeled to consider the shape of the receptive fields.

The experiments indicate that the evaluation matrices based on classification or reconstruction accuracy are insufficient for DL models. The interpretation of the concepts learned and the relevance assigned to the input features for each crop provides insight into the meaningfulness and physical significance of the features learned by the network. VCapsNet fares well in terms of interpretability compared to the other DL models considered in this study. In addition to using the interpretability evaluation for mere quantitative comparison, the same approach can be used to refine the training data and select the most relevant features.

Due to its generalizability, VCapsNet is applicable to crop mapping at different scales. Sub-national and national level mapping requires the generalization of the phenological index curves. The pixel-level phenological VI curves are generalized to yield the crop fingerprints at the field level. The field level VI curves are then classified using VCapsNet to accomplish mapping at different scales. The VCapsNet is generic and can be extended to different applications.

### 4.2. Interpretability Based Comparison of the Benchmark DL Models

The analysis of phenological curves requires appropriate modeling of phenological events in terms of their characteristics such as depth, width, shape, and position. The interpretability-based evaluation strategies (Section 2.2.4) were found to be useful in the understanding of the concepts learned by the models. Comparing the concepts learned for each of the crops with the ancillary data (related to the sowing, growth, and harvesting time) provides an understanding of the learning capability of the DL models. In addition to evaluating the learning capability of the network, the learned concepts also provide an indication of the suitability of the training data and the need to refine this data.

Although analysis of the prototypes learned by DL models provides insights into the learning capability of the network, analysis of the relevance of the different input features is required to interpret the manifold accordingly. As discussed in Section 2.2.4, the LRP approach assigns relevance scores to the input features based on their contributions. Analysis of these relevance scores with respect to the ancillary data regarding the timing of crop cultivation and growth provides an insight into the physical significance of the features learned by the model.

Barley and wheat crops are often indistinguishable from each other and cause issues for crop mapping algorithms. As can be observed from the results, VCapsNet yields good results in classifying these crops. The performance of VCapsNet can be attributed to its ability to appropriately model the crop-specific characteristics of the index curves. The interpretability-based evaluation indicates that the crop-specific phenological events, such as growth transitions, planting, heading, and harvesting, are effectively identified using VCapsNet.

### 5. Conclusions

The VI curves derived from multi-temporal satellite images yield better results for crop classification than the use of single-date image spectra. The ability to model the phenological events and their characteristics determines the effectiveness of the VI curve-based classifiers. However, most of the existing approaches that model the phenological events require manual fine-tuning and are supervised. Moreover, they are prone to outliers and noise. In this regard, this study proposes using a capsule network and variational encoding to model the phenological events and their features. In addition, the approach also allows joint optimization of classification and denoising, thereby solving the issues of outliers and noise. Deep capsule networks are needed to model the complex features of the VI

curves. The current study illustrates that the implementation of higher-level capsules using maxpooling yields better results than the stacked implementation, particularly when the training samples are limited. The routing mechanism is modified to consider the contextual nature of the phenological events. The proposed variational capsule network, called VCapsNet, uses the activity vector of class capsules and the LSTM cell state as information priors to sample the latent space. It was observed that the regularization of classification using the reconstruction loss and that of denoising using the label information priors improve the generalizability and convergence of the network. The constraints and encoding schemes, and the DTW-based similarity measures, achieve effective classification with minimal training samples. Unlike conventional convolution, a point-based convolution is proposed to have flexible receptive fields for data imputation. The proposed strategies ensure the network models the phenological events and characteristic features, and facilitate the use of the information priors to improve the physical significance. Experiments indicate that VCapsNet yields better results than the existing classification and denoising approaches for VI curves. In addition to the confusion-matrix-based accuracy measures, interpretability-based evaluation measures were also employed to analyze the physical significance of the learned features. The predictor-conditioned distribution of the input was modeled to understand the most likely input of the model for a given output. The analysis of these learned concepts allows evaluation of how well the proposed approaches model the phenological events of different crops. In addition, an analysis of the relevance assigned by the model to the input features also helps to understand the learnability of the network. The interpretability-based evaluation measures further facilitate the refinement of the training sets. It was observed that the improvement in interpretability achieved in VCapsNet significantly reduces the sensitivity of the network towards hyper-parameters. The discussed framework and approaches are generic and can also be extended to other applications.

**Author Contributions:** The contributions of the authors include: Conceptualization, P.V.A. and A.K.; methodology, P.V.A.; software, P.V.A.; validation, P.V.A. and A.K.; formal analysis, P.V.A.; investigation, P.V.A. and A.K.; resources, A.K.; data curation, P.V.A.; writing—original draft preparation, P.V.A.; writing—review and editing, P.V.A. and A.K.; visualization, P.V.A. and A.K.; supervision, A.K.; project administration, A.K.; funding acquisition, A.K. Both authors have read and agreed to the published version of the manuscript.

**Funding:** The project was partially funded by Israeli French High Council for Scientific & Technological Cooperation, research program "Maïmonide-Israel", contract no. 3-15832.

**Data Availability Statement:** The data presented in this study are available on request from the corresponding author. The data are not publicly available as the research is currently going on.

**Conflicts of Interest:** No conflict of interest.

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
