# Peer review of "Deep Learning-Based Phenological Event Modeling for Classification of Crops"

_remotesensing, doi:10.3390/rs13132477_

Round 1
Reviewer 1 Report
The authors classify crops by applying a novel methodology based on capsule networks on vegetation index data. They demonstrate that their methodology outperforms the current state-of-the-art methods in terms of accuracy and explainability and does not require any manual tuning. The study is methodologically solid and could benefit from a few modifications to make it more accessible for a remote sensing audience. A minor revision is suggested based on the following points:
Abstract: it would be useful to add another sentence or two at the end of the abstract highlighting the relevance and importance of the results for crop type classification
Methods & Materials
Please consider adding a map showing the distribution of the various fields. Add information on field sizes and the approximate planting and harvest dates for each crop.
It seems like multiple samples are drawn from each field (overall 4600 samples from < 100 fields). While this seems like an efficient way to increase training data size, what is the impact of spatial autocorrelation on the results?
It is not very clear what sort of phenological events are being identified. Is it specific growth stages like V2, V4, etc., or phenological transition dates as identified by curvature-change rate analysis techniques?
To improve an intuitive understanding of the results, I recommend considering adding a time-series graphic that illustrates the efficacy of the VCapsNet based approach in identifying phenological events. This graphic could have time on the X-axis, NDVI on the y-axis, and vertical lines indicating the observed and estimated phenological events. This would complement tables 5 and 6 of the manuscript.
In terms of the validation it may be worth considering to also conduct a temporal validation, i.e. train the model based on 2017-2019 and test on 2020 t and then to repeat the process of 'leave one out' for the other years to further assess the model’s generalizability
Numbering: Section 3.2.1 is repeated 4 times
Table 1: The model names can be simplified by removing the phrase 'Implementation of'. This will increase the readability of this table
Figure 5-6: Since a 10-fold validation is performed for each sub-experiment, can you add the variation in values from the 10-fold validation to the graphs?
Section 3.2.4 Transparency and Explainability: This section can be improved by adding information on the basic intuition behind how explainability is being defined in this study i.e. capturing the importance of when different phenological events occur.
Results/Discussion: Barley and Wheat are often indistinguishable from each other and cause issues for crop mapping algorithms. Please highlight your model performance in separating these two crops and discuss its implications.
Consider adding an image example of the augmented data patterns that were produced.
Can you discuss the scalability and feasibility for i.e. applying this approach for wall to wall crop type mapping at sub-national to national level.
Author Response
File attached.

Reviewer 2 Report
The paper is well written and presented. However, there are some minor comments that need addressing as shown below:
First: format and layout issues
The paper needs a carefully proof reading, as there are several typos and mistakes that can be fixed.
Below are some examples:
L12: cuurent
L122: VI curve classifier is proposed (add a verb),
L124:….. and classification stages is formulated (add a verb)
L180 after the word layer, add respectively
Throughout all the paper, you sometimes use the math expression for the variable names inside the text and sometimes you do not use it, I suggest to be consistent.
In some cases, the figures caption are not directly under the figure (like Fig.1, the caption appeared in a new page).
subsection titles. 3.2.1 appears at the end of the page (move it to a new page)
I suggest moving Fig.1 closer to the text where you have mentioned it for the first time.
L224: length isà are
L262: denotes
Technical issues:
The paper needs more clarifications on some technical issues as discussed below:
Eq. 1, please define the used norm (|| ||)
L180 and 181: can you please clarify the sentence
L188: can you the math symbol you used for the variable v which was mentioned in Eq. 2.
L190: can you give an example for the mentioned interpolation function.
L200/201: can you explain why that is needed/BETTER
L283: corresponds
L378: Kappa not kappa
In some cases, the authors used different referencing style, please be consistent. E.g. L401, 402
Fig. 5 caption is way too long!
L243 and 244 no need to redefine the variables that you have already define before (l234) and 235.
In Eq 8 and 9: can you please clarify what do you mean by || ?
L272: what is 1{.}? can you explain it more
In general, the paper is clear as well as the contribution, especially since the authors compared with other related work and the native DL algorithm.
Author Response
Attached file.

Reviewer 3 Report
The reviewer supports the publication of the manuscript after the correction of the deficiencies observed.
Section "Introduction":
- Lines 74-77. Missing reference.
- Line 80. Reference [35] is not suitable for this context.
- Line 81. The reference Zhong et al. [36] does not seem to support the claim made by the authors in their interpretation of the results. In particular, with regards to the performance of the LSTM model.
I am confused about the intention behind the references presented for lines 84-87. It either requires some rewording or appropriate references. I am assuming that by "irregularly sampled data", the authors were talking about data sampling during the acquisition process (sensor limitations, errors, etc.) and relating it to the inflexible inputs CNNs expect. That's not properly clarified.
- Line 87. Reference [37] is neither related to the generalization capability of CNNs with regards to the amount of data available nor belongs to the remote sensing field, in particular. If the authors consider it relevant, I would appreciate it if they could provide me with an explanation about how the domain-specific problems described in it can be used to draw this conclusion.
- Line 87. Reference [38], again, mentions the problems related to datasets not containing samples for all possible real-world conditions, including some that may be unknown. In my opinion, that problem is not the same as having "a limited number of training samples", as written by the authors.
- Line 87. Reference [39] treats datasets that may be considered as having a low number of training samples but there are no conclusions that can be used to support the statement "fail to effectively process irregularly sampled point data with a limited number of training samples". Please, either narrow the scope of the chosen references to articles related to time series data, select those that specifically treat the problem of having a limited number of training samples, or provide the reasoning behind this particular choice.
- Line 87. Reference [40] is correct.
- Line 87. Reference [40] is not suitable.
- The introduction does not mention either variational autoencoders or capsule networks, contains no references or explanation about them, and includes no citations for other works in the field where they are used. For example https://doi.org/10.1109/TGRS.2021.3058782. Please, adapt the introduction accordingly.
In section "Materials and methods":
- Lines 140-140. Relevant data preprocessing techniques do not constitute a description of the dataset and should be placed in the appropriate section along with information about the parameters if considered relevant to facilitate reproducibility.
In section "Implementation of the Proposed VCapsNet":
- Are there no dropouts?
- What optimizer is being used?
- Please, add information about the variational autoencoder
In section "Results":
- Figure 4. Is there any specific reason for the gaps in the graphs?
- No details are being given of the train/test splits. There is information about the percentage of training samples so the percentage of testing samples can be assumed, but it should be clarified.
- Figure 5. The explanation in the caption of this figure is excessively long and details information about a different table that should either be moved to the text or the corresponding table.
- Due to the previous point, Table 4 is not referenced in the text.
In section "Conclussion":
- Line 579. Does the data from the experimental results prove the requirement of training samples has been reduced to a large extent?
Minor formatting defects:
- Line 90, the fragment starting with "uses one to one.." has a slightly different text color/style.
- Figure 2, colors in the legend do not match the figure for Convolution + Maxpooling.
- Sections 2.1 and 2.2 are mistakenly numbered as 3.1 and 3.2, respectively.
- Section 3.1 is mistakenly numbered as 4.1.
- The numbering of most sections is wrong, I won't list all of the issues in detail. Should be corrected
- After line 393 there is a footnote that does not seem to belong there.
- There is a footnote in the middle of Figure 5 that doesn't seem to belong there.
- Styles of figures 5 and 6 are not consistent. Please review and fix if required.
Author Response
Attached file.
